# Single serine on TSC2 exerts biased control over mTORC1 activation mediated by ERK1/2 but not Akt

Brittany L Dunkerly-Eyring[1,2], Shi Pan[1], Miguel Pinilla-Vera[1], Desirae McKoy[1], Sumita Mishra[1], Maria I Grajeda Martinez[1], Christian U Oeing[1], Mark J Ranek[1,2], David A Kass[1,2]

Tuberous sclerosis complex-2 (TSC2) negatively regulates mammalian target of rapamycin complex 1 (mTORC1), and its activity is reduced by protein kinase B (Akt) and extracellular response kinase (ERK1/2) phosphorylation to activate mTORC1. Serine 1364 (human) on TSC2 bidirectionally modifies mTORC1 activation by pathological growth factors or hemodynamic stress but has no impact on resting activity. We now show this modification biases to ERK1/2 but not Akt-dependent TSC2-mTORC1 activation. Endothelin-1–stimulated mTORC1 requires ERK1/2 activation and is bidirectionally modified by phospho-mimetic (S1364E) or phospho-silenced (S1364A) mutations. However, mTORC1 activation by Akt-dependent stimuli (insulin or PDGF) is unaltered by S1364 modification. Thrombin stimulates both pathways, yet only the ERK1/2 component is modulated by S1364. S1364 also has negligible impact on mTORC1 regulation by energy or nutrient status. In vivo, diet-induced obesity, diabetes, and fatty liver couple to Akt activation and are also unaltered by TSC2 S1364 mutations. This contrasts to prior reports showing a marked impact of both on pathological pressure-stress. Thus, S1364 provides ERK1/2-selective mTORC1 control and a genetic means to modify pathological versus physiological mTOR stimuli.

## Introduction

The mechanistic target of rapamycin complex 1 (mTORC1) controls cell growth, synthetic activity, metabolism, and protein homeostasis (Valvezan & Manning, 2019; Liu & Sabatini, 2020). MTORC1 is activated upon GTP binding to the G-protein Rheb (Ras homolog enriched in brain) which is in turn negatively controlled by the GTPase-activating protein tuberin (tuberous sclerosis complex-2, TSC2) (Huang & Manning, 2008). TSC2 complexes with TSC1, and their bi-directional influence over mTORC1 activity is mediated by TSC2 phosphorylation via multiple serine/threonine kinases (Huang & Manning, 2008). Prominent kinases known to induce constitutive

TSC inhibition to activate mTORC1 include Akt (Inoki et al, 2002; Menon et al, 2014) and extracellular response kinase (ERK1/2) (Ma et al, 2005), each altering multiple serines (five for Akt, three for ERK1/2) mostly in amino acids 500–1,150 of TSC2. Kinases activating TSC to inhibit mTORC1 are AMP activated protein kinase (AMPK), glycogen synthase kinase 3β (GSK3β) (Inoki et al, 2006), and cGMP-stimulated protein kinase G (PKG) (Ranek et al, 2019), modifying amino acids in the 1,300–1,450 range.

Given the multiplicity of signaling kinase inputs that rarely occur in isolation, their interactions and hierarchy is important to understand, yet this remains under studied. A dominant mechanism of TSC1/TSC2 regulation involves its intracellular localization to or away from the lysosomal membrane where Rheb and mTORC1 reside (Menon et al, 2014; Carroll et al, 2016; Demetriades et al, 2016; Fitzian et al, 2021). Nutrient starvation or depletion of growth factors sends the TSC complex to lysosomes in a TSC1-dependent manner (Fitzian et al, 2021) to inhibit mTORC1, whereas growth factor stimulation as by insulin stimulates mTORC1 via translocating the TSC1/2 complex away from the lysosome (Menon et al, 2014). This translocation can be out-competed by nutrient (Menon et al, 2014) or energy depletion (Inoki et al, 2003; Ranek et al, 2019), reducing mTORC1 activity despite growth stimulation. However, little is known about whether and how simultaneous phosphorylation inputs into TSC2 interact, and if they also display signaling bias.

One impediment to studying co-regulation by TSC2 modifiers has been that phospho-mimetic or phospho-silenced mutations used to isolate a given input themselves altered basal mTORC1 activity (Inoki et al, 2002, 2006; Manning et al, 2002; Potter et al, 2002; Ma et al, 2005). However, in 2019, we discovered that conserved tandem serines at human S1364 and S1365 (S1365 and 1366 in mouse; human numbering used here) are phosphorylated by protein kinase G in cardiomyocytes and fibroblasts (Ranek et al, 2019) and confer somewhat different TSC2 regulation. If either serine is mutated to alanine (SA, phospho-null) or glutamic acid (SE, phospho-mimetic), basal mTORC1 activity remains unchanged in vitro and in vivo. Yet if cells (or hearts) expressing these mutant TSC2 are co-stimulated by a $G_{q,11}$-GPCR agonist such as endothelin-1 (ET-1), or cardiac pressure-overload, mTORC1 activity and associated pathological

[1]Division of Cardiology, Department of Medicine, The Johns Hopkins University School of Medicine, Baltimore, MD, USA   [2]Department of Pharmacology and Molecular Sciences, The Johns Hopkins University School of Medicine, Baltimore, MD, USA

Correspondence: dkass@jhmi.edu

growth and disease are markedly amplified when a S1364A mutation is expressed and equally markedly attenuated with a S1364E mutation (Ranek et al, 2019). This control is analogous to a rheostat that sets a gain on a secondary mTORC1 stimulation signal.

Growth factor stimulation of mTORC1 coupled to protein kinase C and mitogen activated kinases driven by pathological stress often includes ERK1/2 signaling, whereas Akt activation is more prominently coupled to insulin and physiologic growth factors (Dibble & Cantley, 2015; Manning & Toker, 2017). Given that TSC2 S1364 is phosphorylated by PKG (Ranek et al, 2019) and that PKG activation counters cardiac pathological but not physiological stress (Takimoto et al, 2009), we speculated that S1364 modulation may have less impact on mTORC1 when stimulated by Akt. We tested this hypothesis by contrasting S1364 phospho-mutation impact on ERK1/2 versus Akt-dependent mTORC1 activation. Here we show S1364 bidirectionally regulates mTORC1 when activated by ERK1/2 but not Akt stimulation. This bias is paralleled by differences in intrinsic S1364 phosphorylation. Akt-mTORC1 signaling potently contributes to diet-induced obesity with diabetes and liver steatosis (Khamzina et al, 2005; Dann et al, 2007; Zhang et al, 2009), and we find these effects are unaltered in homozygous TSC2 knock-in mice expressing S1364A or S1364E. Last, S1364 minimally alters mTORC1 modulation by energy or nutrient supply. Thus, TSC2 S1364 provides bi-directional modulation of mTORC1 activation biased to ERK1/2 but not to Akt or nutrient signaling.

## Results

### TSC2 S1364 bidirectionally regulates ERK1/2 activation of S6K

ET-1 interacts with both an A- and B-type receptor, the former being more prominent in cardiomyocytes and coupled to $G_{\alpha q}$-signaling activating ERK1/2, the latter in vascular cells and linked to PI3K and Akt signaling (Davenport et al, 2016). We therefore tested the role of ERK1/2 activation with ET-1 stimulation in rat ventricular myocytes (NRVM). Myocytes exposed to ET-1 exhibited increased S6K (S389) phosphorylation that was dose-dependently blocked by co-incubation with the ERK1/2 inhibitor SCH772984 (SCH; 0.01–10 $\mu$M; Fig 1A) and another selective inhibitor U0126 (Fig S1A). This was paralleled by altered ERK1/2 activity as reflected by RSK phosphorylation (Figs 1A and S1A). U0126 but not SCH reduced ERK1/2 phosphorylation itself in these cells. SCH772984 dose-dependently increased Akt phosphorylation (pT308) that might reflect a feedback signal, but did not prevent p/t S6K from falling to control levels. Conversely, we found no impact of Akt inhibition (MK2206: MK - 0.15–15 $\mu$M, Fig 1B) on ET-1–stimulated S6K.

We next tested the consequence of expressing TSC2 S1364 phospho-mutants–S1364E and S1364A on ET-1 responses. ET-1–stimulated p/t S6K dose-dependently declined with increasing TSC2$^{S1364E}$ expression (Fig 1C), despite a rise in p/t Akt. ERK1/2 activation (Fig 1C) and RKS2 activation (Fig S1B) were unaltered by TSC2$^{S1364E}$ expression as these were upstream of the TSC2 mutation. Co-incubation with SCH did not further depress p/t S6K in cells expressing TSC2$^{S1364E}$, whereas p/t RSK2 was reduced significantly by SCH or U0126 (Fig S1B).

Myocytes expressing the phospho-silenced mutation TSC2$^{S1364A}$ exhibited the opposite behavior, with ET-1–stimulated p/t S6K increasing more despite similar ERK1/2 activation. S6K activation and its enhancement by TSC2$^{S1364A}$ were fully blocked with co-exposure to either ERK1/2 (SCH772984) or mTORC1 (rapamycin) inhibition (Fig 2A and B). Our prior report (Ranek et al, 2019) showed analogous changes with each S1364 mutant on ET-1 stimulation of 4EBP-1 and Ulk1, so here we just focused on S6K phosphorylation.

Our earlier study (Ranek et al, 2019) had found that by activating PKG with a phosphodiesterase type 5 (PDE5) inhibitor, in vitro S6K activation by ET-1 or in vivo by pressure overload was suppressed. This did not occur, however, if TSC2 S1364 phosphorylation was prevented by expressing the S1364A mutation. In a subsequent study, we revealed this is specific to PDE5 inhibition. Stimulating PKG by either blocking PDE9 (the other highly cGMP-selective PDE) or by activating guanylyl cyclase-1 (GC-1) similarly depressed ET-1–stimulated p/t S6K, but this occurred even in cells expressing TSC2$^{S1364A}$ (Oeing et al, 2020). This disparity was particularly notable in myocytes co-expressing a non-oxidizable form of PKG1$\alpha$ (PKG1$\alpha^{C42S}$) that enhances its localization to the plasma membrane and in turn regulation of S1364. One way to explain these results is that unlike PDE5 inhibition, the two other modes of PKG activation also suppress ERK1/2 activation proximal to TSC2. If so, then our current data indicate this should circumvent a S1364A mutation. We tested this hypothesis in NRVMs expressing both TSC2$^{S1364A}$ and PKG1$\alpha^{C42S}$ (Fig 2C). ET-1–stimulated ERK1/2 was indeed unaltered by PDE5 inhibition yet significantly reduced by either PDE9 inhibition or GC-1 stimulation. These differences in ERK1/2 activation correlated with changes in p/t S6K, supporting our hypothesis. These results further highlight the critical engagement of ERK1/2 stimulation for S1364 regulatory control of mTORC1.

Fig 2D plots the relation between p/t ERK1/2 or p/t Akt and p/t S6K derived from the studies reported in Figs 1B and 2C. To combine these gels, the data were bi-normalized to a mean of 1.0 for vehicle control and peak change of 4.0 (e.g., fourfold increase over vehicle). There is a strong positive correlation between p/t S6K and p/t ERK1/2 but no relation with p/t Akt. These studies identify ERK1/2 is a major S6K co-stimulant that is bidirectionally modified by TSC2 S1364E or S1364A modification.

### TSC2 S1364 minimally regulates Akt-stimulated S6K

Physiological growth factors such as insulin are major activators of mTORC1 via Akt phosphorylation of TSC2. Therefore, we tested whether such signaling is also regulated by TSC2-S1364, examining both insulin and another Akt-coupled signaling peptide - platelet derived growth factor (PDGF). Insulin-activated Akt and S6K in MEFs, co-activating ERK1/2 as well (Mohan et al, 2017) but to a much lesser extent (Fig 3A). Importantly, inhibiting ERK1/2 did not alter the rise in p/t S6K, whereas this was blocked by Akt inhibition, suggesting the latter was the primary stimulus. The insulin response was then studied in TSC2 KO MEFs infected with adenovirus expressing either an empty vector, or TSC2$^{WT}$, TSC2$^{S1364E}$, or TSC2S$^{S1364A}$. KO cells exhibited greater basal p/t S6K, and this was reduced by expression of any of the TSC2 forms. Insulin activated S6K, but this was unaltered by TSC2$^{S1364E}$ (Fig 3B). This contrasts with full suppression of insulin-activated S6K with mTORC1 inhibition (rapamycin or torkinib, Fig 3C). Cells expressing TSC2$^{S1364A}$ displayed a modest rise in

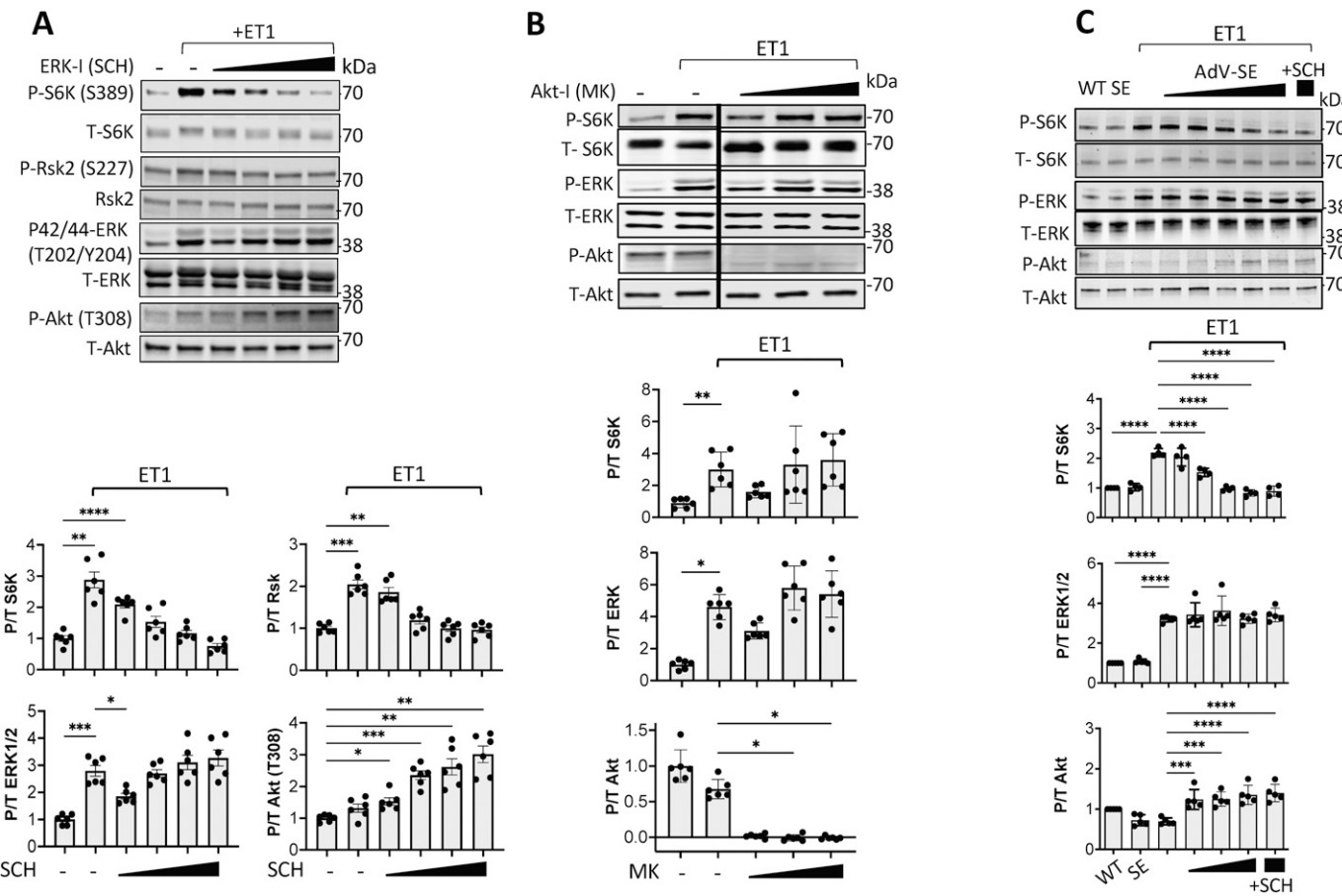

**Figure 1. ET-1 stimulated S6K by ERK1/2 is suppressed by selective ERK1/2 but not Akt Inhibition and by TSC2-S1364E.**
**(A)** Example immunoblots and summary data of dose dependent decline in ET-1 induced S6K and RSK2 phosphorylation by ERK1/2 inhibition (SCH) in NRVMs. Concomitant ERK1/2 phosphorylation itself did not decline, whereas Akt phosphorylation rose with increasing SCH dose (0.01–10 μM). N = 6/group; ****$P < 1 \times 10^{-4}$, ***$P \leq 0.0006$, **$P \leq 0.006$, *$P = 0.01$; Welch one-way ANOVA, Dunnett's multiple comparisons test (MCT). **(B)** Inhibition of Akt does not significantly reduce p/t S6K stimulated by ET-1. N = 6/group; Kruskal–Wallis test $P < 0.0007$ for each; Dunn's MCT: **$P = 0.002$; *$P < 0.026$. **(C)** Expression of TSC2$^{S1364E}$ (SE) reduces ET-1–stimulated S6K in a dose-dependent manner; ERK1/2 activity is unchanged, and Akt activity rises with higher SE expression. N = 4/group; one-way ANOVA ($P \leq 3 \times 10^{-10}$ for each protein), Sidak's MCT: ****$P < 10^{-5}$; ***$P < 0.0002$. Source data are available for this figure.

p/t S6K over WT perhaps from co-activated ERK1/2. In each condition, p/t Akt and TSC2 expression were similar. Insulin also stimulated 4EBP-1, though this was not amplified in cells expressing TSC2$^{S1364A}$ nor reduced in those expressing TSC2$^{S1364E}$ (Fig S2A). We found a similar lack of impact from S1364 modulation on the S6K and 4E-BP1 responses to PDGF (20 ng/ml × 30 min). PDGF induced robust p/t Akt and p/t S6K, yet both were unaltered despite expression of TSC2$^{S1364A}$ or TSC2$^{S1364E}$ mutants (Figs 3D and S2B). Thus, unlike ERK/12-mediated signaling, mTORC1 activation by Akt-dominant pathways is little modified by S1364 TSC2.

### Influence of S1364 modulation on balanced ERK1/2 and Akt co-stimulation

As the preceding results supported biased TSC2 S1364 regulation of ERK1/2 but not Akt-stimulated mTORC1, we examined responses to thrombin that co-stimulates each kinase (Fig 4A). The corresponding rise in p/t S6K is TSC2 dependent, as it was absent in TSC2-KO MEFs but restored by re-expressing WT-TSC2 (Fig 4B). Akt

blockade reduced thrombin-stimulated p/t S6K only in WT but not TSC2-KO MEFs, leaving p/t ERK1/2 unchanged. Blocking ERK1/2 in thrombin-stimulated KO MEFs that re-expressed TSC2-WT reduced p/t S6K without altering p/t Akt (Fig 4C), further supporting contributions from both pathways. MEFs expressing TSC2$^{S1364E}$ had reduced p/t S6K overall (Fig 4D and E), and whereas this did not alter reduction from Akt inhibition (Fig 4F), it blunted a decline with co-ERK1/2 inhibition (Figs 4G and S3; $P = 0.02$ for genotype-ERK1/2 inhibition interaction). A small but significant increase in p/t S6K from thrombin in MEFs with TSC2$^{WT}$ or TSC2$^{S1364E}$ and co-ERK1/2 inhibition was statistically similar and likely reflected unaltered effects from Akt activation (Fig 4G). Thus, in the presence of ERK1/2–Akt co-activation, TSC2 S1364 selectively modifies p/t S6K related to ERK1/2 and not Akt.

### Increased S1364 phosphorylation by ET-1 but not insulin-stimulated mTORC1

Amplification of ET1-ERK1/2 but not insulin-Akt stimulation of mTORC1 in cells expressing TSC2$^{S1364A}$ suggests this serine may be

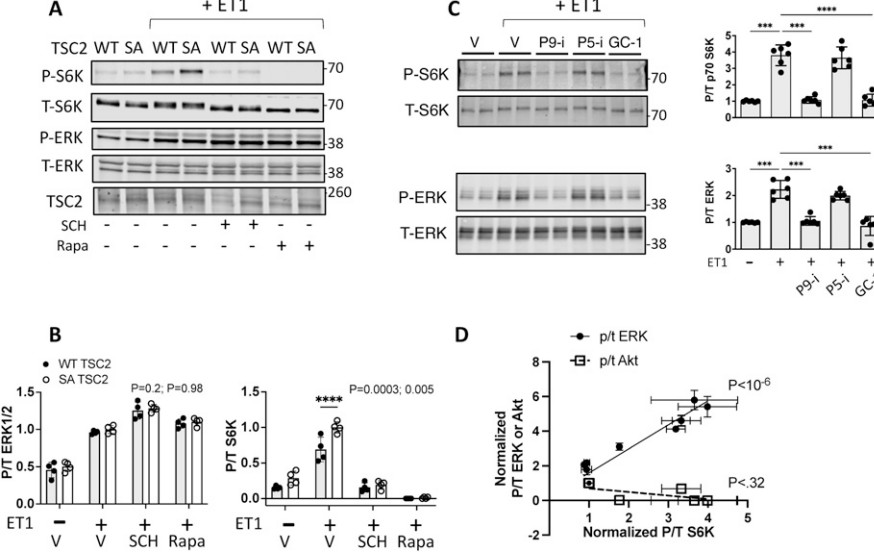

Figure 2. ET-1–stimulated S6K is amplified by TSC2-S1364A and requires ERK1/2 activation.
(A, B) Example immunoblots and (B) summary data showing ET-1–stimulated S6K in NRVMs is augmented by expression of TSC2$^{S1364A}$ (SA) without altering ERK1/2 phosphorylation. This stimulation is blocked by ERK1/2 (SCH) or mTOR (rapamycin, Rapa) inhibition. N = 4/group; P-values for two-way ANOVA for genotype and interaction of genotype and condition. ****P < 0.0001 by Sidak's MCT. (C) Effect of ET-1 stimulation on S6K and ERK1/2 phosphorylation in NRMVs co-expressing TSC2$^{S1364A}$ and PKG1α$^{C42S}$ and then exposed to vehicle (Veh), PDE5 (P5-i) (1 μM), or PDE9 (P9-i) (0.1 μM) inhibition, or soluble guanylate cyclase activation (GC-1) (0.1 μM). N = 6/group; Welch one-way ANOVA, Dunnett's MCT; ***P ≤ 0.001; ****P = 0.00006. (D) Combined analysis of data from Figs 1B and 2C bi-normalized to span from 1.0 with vehicle control to 4.0 for peak p/t S6K response. Data show strong correlation between p/t S6K and p/t ERK1/2 but not p/t Akt. P-values are for linear regression of each respective relation.
Source data are available for this figure.

intrinsically engaged as a counter-brake to mTORC1 with the former but not latter pathway. To test this, we exposed NRVMs to ET-1 or insulin for 15 min, and then incubated cells with either ERK1/2, Akt, PKG (DT3, 1 μM) inhibitors or vehicle control. S1364 phosphorylation increased with ET-1 and was unaltered by co-ERK1/2 inhibition yet blocked by co-PKG inhibition (Fig 4H). In contrast, pS1364 did not increase with insulin nor altered further by any of the kinase inhibitors. Thus, ET-1 but not insulin stimulation results in PKG-dependent S1364 phosphorylation. This supports intrinsic differential regulation involving TSC2 S1364 as a mechanism underlying signaling bias.

### S1364 minimally controls mTORC1 via energy/nutrient depletion/repletion

Energy depletion by incubation with 2-deoxyglucose activates AMPK, and we previously showed this blocked S6K activation equally in ET-1–stimulated TSC2$^{WT}$ and TSC2$^{S1364A}$ expressing KO MEFs (Ranek et al, 2019). This supports over-riding control over S1364 modulation by energy supply. Here we further tested this using a more pathophysiological model involving simulated ischemia (SI) to activate AMPK (Mungai et al, 2011). This also markedly reduced p/t S6K and p/t 4E-BP1 independent of the TSC2 S1364 mutation being expressed (Figs 5A and S2C).

Depletion or abundance of growth factors and/or nutrients is a potent modulator of mTORC1 activity (Menon et al, 2014; Demetriades et al, 2016). To test if TSC2 S1364 mutations regulate these stimuli, NRVMs expressing WT, S1364A, or S1364E TSC2 were cultured in FBS followed by serum-free conditions (each 1 h). Cells in FBS expressing TSC2$^{S1364A}$ had slightly higher p/t S6K (Fig 5B) consistent with the presence of growth factors–some engaging ERK1/2. However, S6K activity declined markedly upon serum depletion independent of the TSC2 genotype. We also tested the effects of amino acid depletion/repletion, with cells placed in dialyzed FBS lacking amino acids and then switched to media

containing them (each 1 h). The ratio of p/t S6K and p/t 4E-BP1 were very low with amino acid depletion and rose similarly in all TSC2 genotype groups with repletion (Figs 5C and S2D). Together, these data indicate that TSC2 1364 modification does not influence energy or nutrient stimuli associated with S6K or 4EBP-1 activation or inhibition.

### S1364 phosphosite mutagenesis does not alter diet-induced obesity

In hearts subjected to pressure-overload or ischemia-reperfusion injury, the mutational status of TSC2 (e.g., TSC2$^{S1364A}$ or TSC2$^{S1364E}$) potently altered mTORC1 activation and the corresponding pathophysiology (Ranek et al, 2019; Oeing et al, 2021). Both conditions stimulate ERK1/2, the former along with activation of multiple pathogenic kinases (Bueno et al, 2000), the latter as a protective mechanism (Lips et al, 2004). By contrast, diet-induced obesity stimulates mTORC1 in fat, liver, and skeletal muscle (Khamzina et al, 2005) primarily by Akt activation (Khamzina et al, 2005; Dann et al, 2007; Zhang et al, 2009). Given the lack of regulation of Akt-mTORC1 stimuli by S1364 in the current cellular studies, we hypothesized that TSC2$^{WT}$, TSC2$^{S1364A}$, or TSC2$^{S1364E}$ KI mutations would also have minimal impact on diet-induced obesity.

Knock-in mice with homozygous expression of each TSC2 phosphosite mutant and littermate WT controls were subjected to 60% high fat diet for 18 wk. We found each genotype exhibited nearly identical increases in body weight (Fig 6A). Each group also developed type-2 diabetes reflected by elevated fasting plasma glucose and an abnormal glucose tolerance test, and these were also similar between groups (Fig 6B and C). We documented extensive hepatic lipid deposition when compared with standard-diet aged-matched controls (Fig 6D), and this was unaltered by the TSC2 S1364 mutations. Thus, in a model known to engage Akt-mediated mTORC1 activation as a major component of its pathophysiology, TSC2$^{S1364A}$ or TSC2$^{S1364E}$ mutations had no impact.

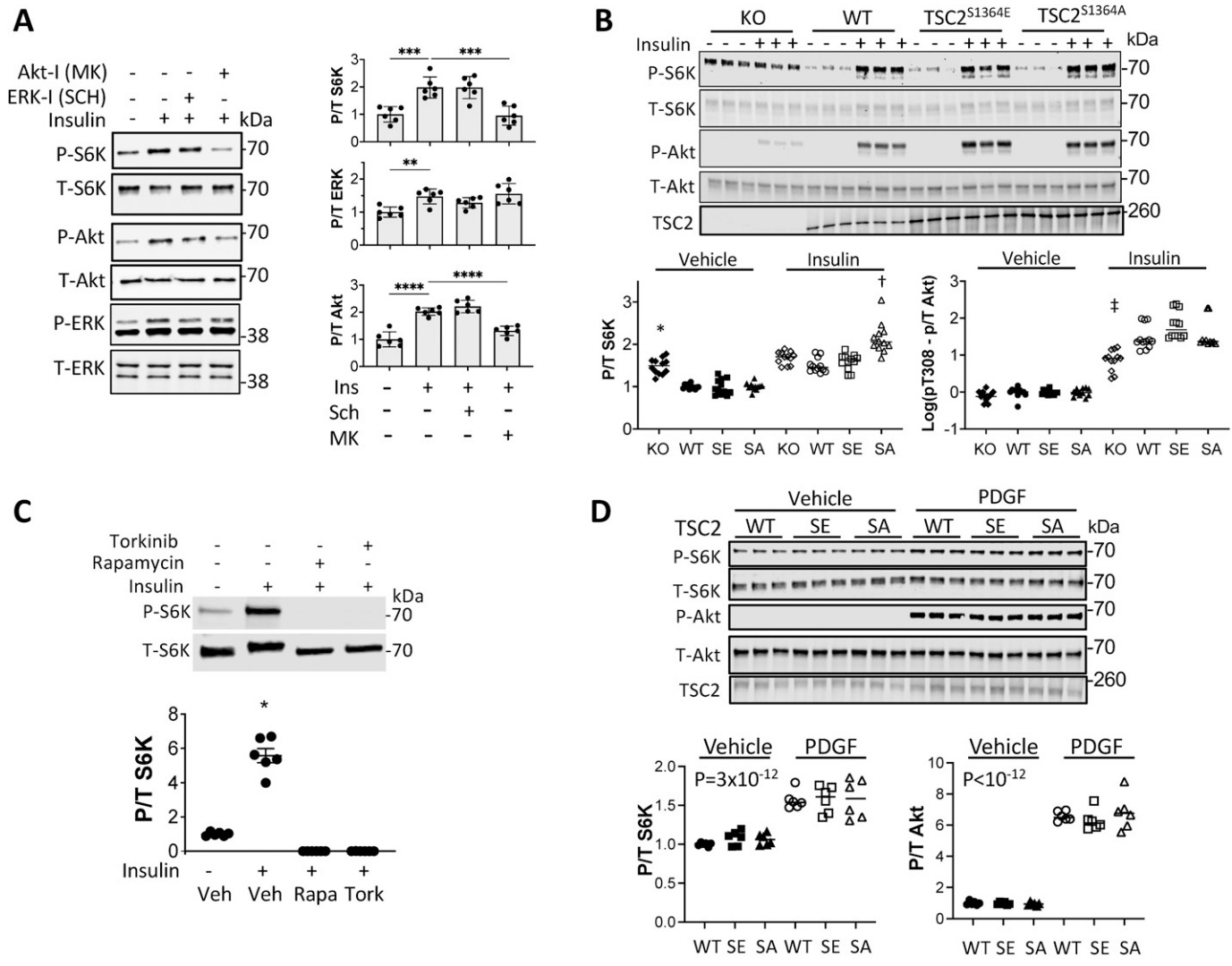

**Figure 3. S1364 modulation does not regulate Akt-stimulated pathways.**
**(A)** Insulin-stimulated pS6K in MEFs is not altered by ERK1/2 inhibition but is by Akt inhibition; example immunoblot (left) and summary data (right) shown. N = 6/group; one-way ANOVA, Holm–Sidak MCT: **$P \leq 0.004$, ***$P = 0.0002$; ****$P \leq 2 \times 10^{-5}$. **(B)** Example immunoblot and summary results for insulin stimulation in TSC2 KO MEFs infected with AdV expressing empty vector (KO), or WT, SA, or SE TSC2. N = 12/group; two-Way ANOVA, Sidak's MCT; *$P < 2 \times 10^{-7}$ versus Vehicle KO; †$P < 3 \times 10^{-7}$ versus KO+ insulin; ‡$P \leq 6 \times 10^{-7}$ versus KO+ insulin. **(C)** Insulin-stimulated S6K is fully blocked by rapamycin (Rapa) or torkinib (Tork), contrasting to effect from SE mutant. N = 6/ group; Welch one-way ANOVA, Dunnett's test; *$P < 0.0003$ versus other groups. **(D)** PDGF stimulation potently activates Akt and S6K, and this response is not significantly altered by TSC2 mutants versus WT. N = 6/group; two-way ANOVA, P = $3 \times 10^{-12}$ for PDGF effect, 0.7 for genotype effect, and 0.8 for genotype–PDGF interaction. Source data are available for this figure.

This supports our cell-based data showing little regulatory impact of TSC2 S1364 modulation on Akt-dependent signaling.

# Discussion

Modulation of mTORC1 activity by TSC2 involves integrating multiple signaling inputs, the majority being transduced by phosphorylation of specific serines and threonines residing in a ~900 amino acid mid-region of the protein. Unlike Akt, ERK1/2, AMPK, or GSK3β that require altering multiple residues to modulate mTORC1 (Inoki et al, 2002, 2003, 2006; Manning et al, 2002; Ma et al, 2005), modification of

one serine at S1364 is sufficient to confer regulatory effects, although not on basal mTORC1 activity but only upon its co-stimulation. Here, we show this regulation is biased to modulate ERK1/2 but not Akt activating pathways and does not alter mTORC1 modulation by nutrient or amino acid supply.

Fig 7 summarizes the signaling revealed in this study. Whereas both ERK1/2 and Akt activation suppress TSC2 inhibition of mTORC1, only ERK1/2 stimuli are co-regulated by the phospho-status of TSC2 S1364. With a combined stimulus such as thrombin, the ERK1/2 not Akt component is modulated. In vivo, pathology such as diet-induced obesity that prominently engages Akt stimulation is not impacted by the S1364 status, whereas as previously reported, pressure overload that engages ERK1/2 is Ranek et al (2019). There

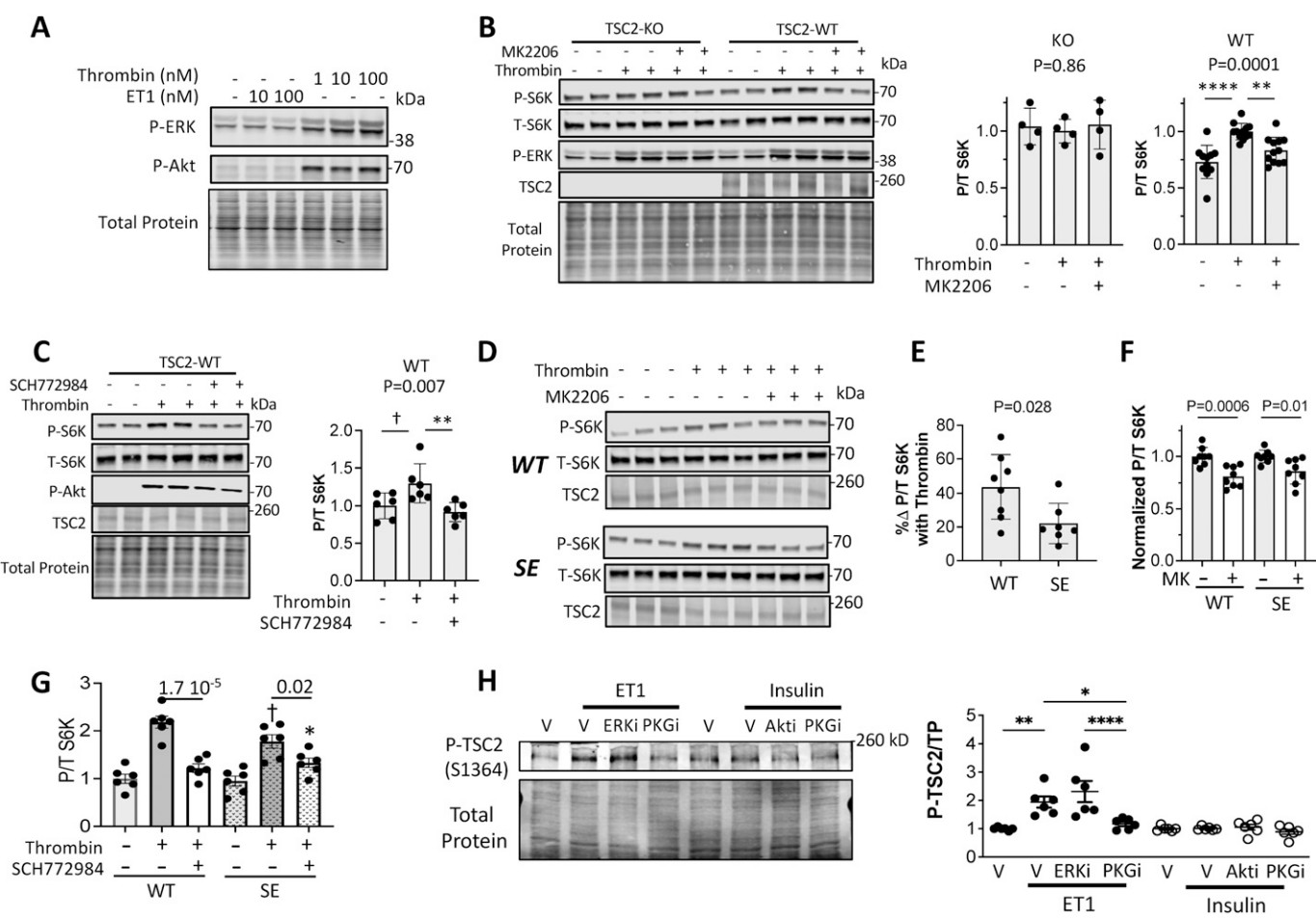

**Figure 4.  Effect of S1364 on ERK1/2 - Akt co-activation and impact of each on pS1364.**
**(A)** Example immunoblot of MEFs exposed to thrombin showing balanced activation of ERK1/2 and Akt. **(B)** Example immunoblot and summary data for TSC2 KO MEFs and cells with WT TSC2 re-expressed, exposed to thrombin ± Akt inhibition (MK2206). N = 4/group TSC2-KO; N = 11/group WT. P-values displayed are for Kruskal–Wallis test, ****P = .00009, **P = 0.009 by Dunn's MCT. **(C)** Example immunoblot and summary data for the same experiment but ± ERK1/2 inhibitor (SCH). N = 6/group; KW test P-values displayed; Dunn's MCT: †P = 0.1, **P = 0.007. **(D)** Example immunoblot for KO MEFs re-expressing either TSC2-WT or TSC2$^{S1364E}$, stimulated with thrombin ± Akt inhibitor (MK2206). **(E)** Percent rise in p/t S6K due to thrombin in WT versus SE TSC2 expressing KO MEFs. N = 7–8/group; P-value displayed Mann–Whitney U test. **(F)** Effect of Akt inhibition on p/t S6K response to thrombin in TSC2-KO MEFs expressing WT or SE TSC2. N = 8/group; two-way ANOVA, Sidak's MCT—P-values displayed. **(G)** Effect of ERK1/2 inhibition by SCH772984 on p/t S6K in TSC2-KO MEFs expressing WT or TSC2$^{S1364E}$. two-way ANOVA with Sidak's MCT P-values shown. †P = 0.04 versus WT; *P = 0.022 versus SE with vehicle controls. **(H)** Differential effect of ET1 or insulin stimulation on TSC2 phosphorylation at S1364 N = 6/group; one-way ANOVA, Sidak's MCT. *P = 0.012; **P = 0.001; ****P = 0.00008.
Source data are available for this figure.

are many receptor tyrosine kinases, cytokines, immune antigen receptors, and G-protein coupled receptors that predominantly stimulate via Akt (Manning & Toker, 2017), and we suspect these are unlikely to be altered by S1364 modulation. There are also many pathways engaging ERK1/2, including GPCRs, oxidant stress, protein kinase C, and RSK stimulation (Ma et al, 2005), and we expect these will be modified by S1364 as well. The new findings identify a novel nuanced approach to mTORC1 regulation.

The strong bias for S1364 to control ERK1/2 but not Akt modulation of mTORC1 is consistent with its intrinsic phosphorylation by the former but not latter stimulus. Gq-coupled GPCRs activate ERK1/2 by phospholipase C$\beta$-PKC and Ras-dependent pathways. Both ET-1 (Kapakos et al, 2010; Nakamura et al, 2015) and insulin (Anfossi et al, 2009; Yu et al, 2011) co-activate nitric oxide synthase dependent cGMP-PKG via Akt-dependent stimulation. Yet we only observed pS1364 with ET-1 stimulation, suggesting S1364 is an intrinsic and selective counter-brake on certain mTORC1 activation pathways but not others, with those that themselves enhance S1364 phosphorylation being impacted. PKG is not the only kinase that can phosphorylate S1364; PKC does this as well (Ballif et al, 2005), and other kinases may play a role in various cell types.

Evidence to date shows that rather than altering TSC2-GTPase activity, the primary mechanism, whereby TSC2 alters mTORC1 signaling is by moving the complex away from or to the lysosomal membrane. As first reported by Manning and Demetriades, starvation and growth factor or amino acid depletion moves TSC2 to the lysosome to reduce Rheb-GTP binding and suppress mTORC1, whereas Akt activation does the opposite (Demetriades et al, 2014; Menon et al, 2014; Demetriades et al, 2016). In 2021, Fitzian et al (2021) reported that TSC1 is central to this translocation, binding to

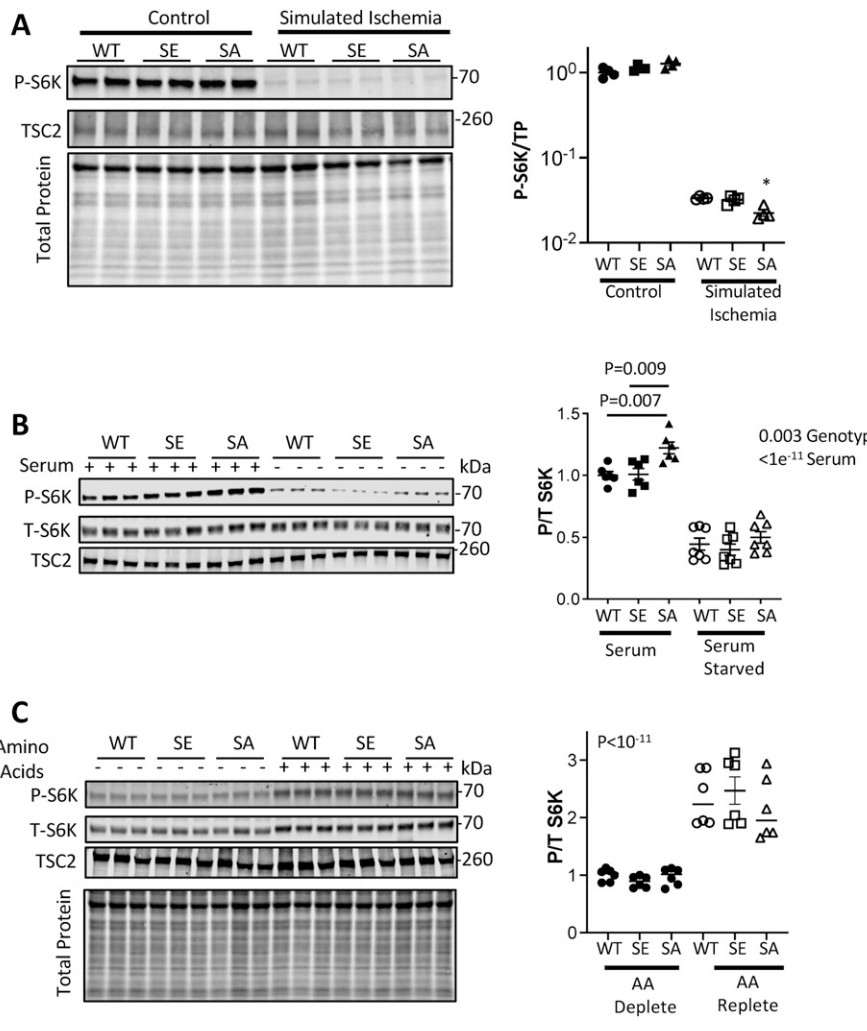

**Figure 5. TSC2 S1364 minimally influences energy or nutrient regulation of S6K.**
**(A)** NRMVs exposed to simulated ischemia (SI) display marked suppression of S6K activation that is similar regardless of the TSC2 S1345 genotype. There is slightly less activation after SI in TSC2$^{S1364A}$ expressing cells. N = 4/group; Kruskal–Wallis test, Dunn's MCT: *$P$ = 0.03 versus WT-SI. **(B)** Effect of serum depletion on S6K activation. p/t S6K is significantly greater in SA expressing cells, but all decline with serum depletion independent of TSC2 genotype. N = 6–7/group; two-way ANOVA $P$-values provided; Sidak's MCT: $P$-values shown. **(C)** Amino acid (AA) repletion increases p/t S6K similarly independent of the TSC2 genotype expressed. $P$-values are for two-way ANOVA genotype effect. Effect of genotype: $P$ = 0.72; Genotype × AA repletion, $P$ = 0.44.
Source data are available for this figure.

phosphoinositol phospholipid PI3,5P2 at the lysosomal membrane to position TSC2 for mTORC1 inhibition. Whether this or other mechanisms apply to TSC complex movement from lysosomes after Akt or ERK1/2 activation or their selective modulation by S1364 remains unknown. However, the lack of basal phenotypes with both TSC2$^{S1364A}$ and TSC2$^{S1364E}$ mutants, and their preservation of nutrient, energy, and physiological growth factor signaling via Akt suggests something else is likely involved. To our knowledge, data showing ERK1/2 activation translocates TSC1/TSC2 to the cytosol has yet to be reported, but if it did, then the TSC2$^{S1364E}$ mutation would seem to reverse it whereas a TSC2$^{S1364A}$ mutation augment such translocation. Mechanisms for such bi-directional and biased control have yet to been determined.

MTORC1 activation in diet-induced obesity (DIO) is considered central to the pathophysiology, contributing to fat deposition in the liver (Wang et al, 2015; Han & Wang, 2018), type-2 diabetes and insulin desensitization (Bar-Tana, 2020), lipogenesis (Zhang et al, 2009), and other abnormalities. Mice lacking S6K are resistant to DIO and metabolic syndrome (Um et al, 2004). Yet, we found none of these abnormalities were impacted by preventing or mimicking TSC2 S1364 phosphorylation. This is strikingly different from the

potent influence from the same TSC2 mutations on modulating pressure-overload stress (Ranek et al, 2019). Prior studies found that PKG activation, which phosphorylates TSC2 S1364 suppresses pressure-overload cardiac hypertrophy yet does not block cardiac hypertrophy caused by myocyte-specific Akt hyperactivation (Takimoto et al, 2005). These in vivo results further support the biased regulation over ERK1/2 versus Akt signaling inputs into TSC2.

Most of the experiments in this study stimulated pathways using receptor ligands as opposed to over-expressing a constitutively active kinase. We chose this approach to generate acute changes rather than have prolonged signaling that could trigger other compensations. Although these ligands do not trigger only one pathway (e.g., ERK1/2 versus Akt), their balance (e.g., ET-1, thrombin, insulin: mostly ERK1/2, both, mostly Akt) were sufficiently different to address the hypothesis. We also used selective kinase inhibitors rather than genetic silencing methods, as the latter may alter other signaling as well. We have not yet identified the intra-molecular mechanism by which TSC2 S1364 selectively modulates ERK1/2 but not Akt signaling. However, recent structural data and ongoing efforts should pave the way for such insights in the future (Yang et al, 2021). The finding of parallel disparities in the phosphorylation

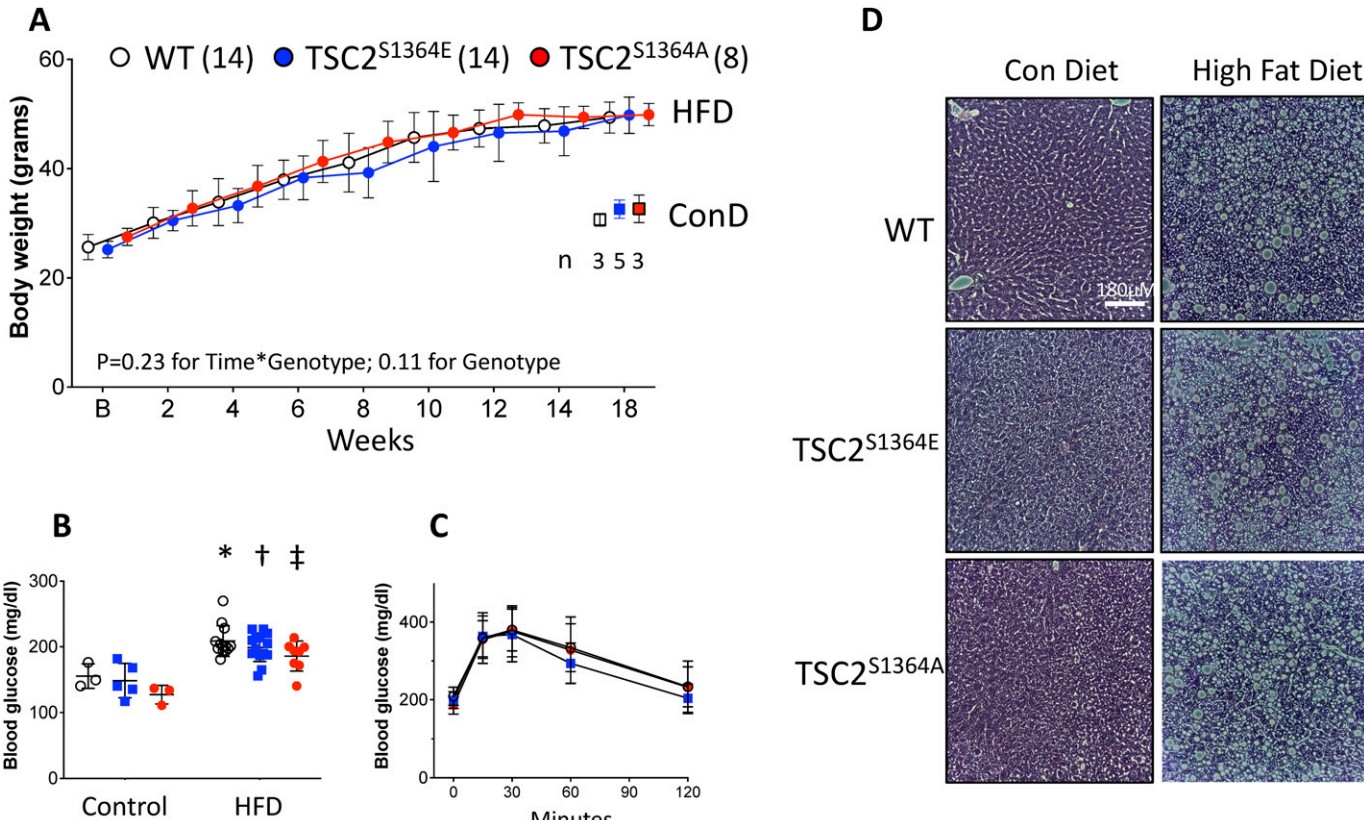

**Figure 6. TSC2 S1364 does not regulate response to diet induced obesity.**
**(A)** Body weight increases similarly in homozygous knock-in mice with TSC2$^{S1364A}$, TSC2$^{S1364E}$, or littermate (WT) controls over an 18-wk duration of high fat diet (HFD). Results for control diet (ConD) at the later time point are also displayed. Group sizes provided in figure. Analysis by two-way ANOVA, $P = 0.23$ for time × genotype; $P = 0.11$ for genotype; $P < 10^{-10}$ for time effect. **(B)** Fasting blood glucose for each TSC2 genotype. Two-way ANOVA: $P = 0.9$ for genotype × diet interaction; HFD versus control diet: *$P = 0.002$, †$P = 0.0003$; ‡$P = 001$. **(C)** Glucose response test for each group. N = 6/group; two-way ANOVA: $P = 0.31$ for time × genotype, 0.57 genotype, $P = 10^{-9}$ time. **(D)** Example H/E-stained liver histology from WT, TSC2$^{S1364A}$, and TSC2$^{S1364E}$ KI mice with control versus HFD. There was similar marked hepatic steatosis regardless of the TSC2 genotype. Controls on standard diet are shown in the left column. Data representative of group results n = 6–8/group, with $P > 0.6$ for genotype effect on steatosis ($\chi^2$ test). Source data are available for this figure.

of S1364 with ET-1 but not insulin stimulation may help identify the mechanism.

To our knowledge, S1364 is the first TSC2 regulatory site revealed that leaves basal mTORC1 signaling intact despite mutagenesis to a phospho-silenced or mimetic form yet confers selective modulation of mTORC1 activity upon co-stimulation with bias to ERK1/2 but not Akt activation. From a translational perspective, the results open opportunities to modulate TSC2-mTORC1 regulation in a more selective manner, with potential applications to cell therapies for cancer and autoimmune disease. From a general perspective, they reveal how phosphorylation of TSC2 can confer selective amplification or attenuation of other kinase inputs via the same protein to control net mTORC1 activity.

## Materials and Methods

### Cell culture models

Neonatal rat ventricular myocytes (NRVMs) were freshly isolated as previously described (Lee et al, 2015) and cultured at 1 million cells per well in six-well plates for 24 h in DMEM with 10% FBS and 1% penicillin/streptomycin to 80–90% confluency before transfection with plasmid or infection with adenovirus. In addition, studies were performed using MEFs and MEFs with TSC2 genetically deleted. They were cultured in DMEM with 10% FBS and 1% pen/strep to 50–60% confluence. Both NRVMs and TSC2 KO MEFs were modified to express TSC2$^{WT}$, TSC2$^{S1364A}$, or TSC2$^{S1364E}$ (WT, SA, or SE) protein using either adenovirus or plasmid expression vectors applied at reported MOI (Oeing et al, 2020) for 3–4 h in serum-free media. In one study, NRVMs were also modified to express PKG1α C42S mutant protein also as previously described (Oeing et al, 2020). Plasmids were applied 24 h after plating (5 μg with Takara Clontech Xfect in NRVMs, 2.5 μg and 7.5 μl Lipofectamine LTX reagent, Thermo Fisher's Lipofectamine LTX & PLUS Reagent, MEFs) following the manufacturer's protocol. All cells were provided 48 h to express the TSC2 proteins before study.

### Pharmaceuticals, antibodies, plasmids, and vectors

The following pharmaceuticals were used: recombinant endothelin-1 (E7764; Sigma-Aldrich), thrombin (HT 1002a; Enzyme Research Laboratories), PDGF-BB (PMG0044; Gibco), ERK1/2 inhibitors (SCH772984,

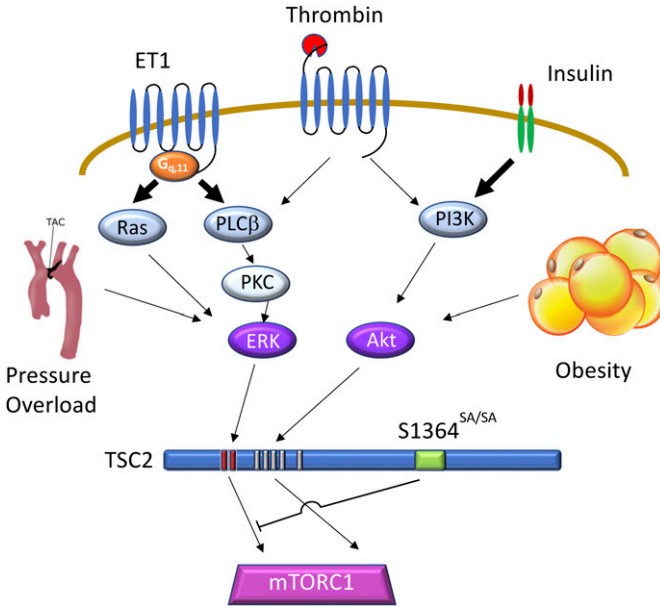

**Figure 7.   Schematic of TSC2-1364 regulated signaling.**
A $G_{q,11}$ – GPCR typified by ET-1 receptor and tyrosine kinase receptor typified by insulin receptor, and mixed receptor (e.g., thrombin) are depicted with their downstream signaling. ET-1 prominently activates PLCβ-PKC-ERK1/2 and Ras-ERK1/2 pathways leading TSC2 phosphorylation reducing its suppression of mTORC1 to increase p/t S6K. Insulin more prominently activates PI3K-Akt, phosphorylating TSC2 at different sites but also leading to mTORC1 activation. Thrombin engages both cascades. TSC2 S1364—modified by mutagenesis to block (S1364A, SA) or mimic (S1364E, SE) its phosphorylation—alters co-modulation of mTORC1 due to ERK1/2 but not to Akt signaling via TSC2. In vivo examples of this bias is found by marked impact of S1364 modulation on cardiac pressure overload as reported (Ranek et al, 2019), but negligible effect on high-fat diet obesity and metabolic syndrome observed in the current study.

#S7101; Selleck Chemical; U0126, #9903; Cell Signaling), Akt inhibitor (MK-2206, S1078; Selleck Chemical), insulin (I9278; Sigma-Aldrich), Torkinib (S2218; Selleck Chemical), Rapamycin (R8781; Sigma-Aldrich), PDE9 inhibitor (PF04447943; Pfizer), PDE5 inhibitor (sildenafil; Pfizer), and soluble guanylate cyclase activator (BAY-602770; Bayer).

Primary antibodies were targeted to: Total TSC2 #4308, phospho-p70 S6 kinase (Thr389) #9205, total p70 S6 kinase #9202, phospho-4E-BP1 (Ser65) #9451 and total 4E-BP1 #9452, phospho-Akt (Ser473) #9271, phospho-Akt (Thr308) #13038, and total Akt #9272, phospho-p44/42 MAPK (ERK1/2) (Thr202/Tyr204) #9101 and total p44/42 MAPK (ERK1/2) #9102, pRSK (T359/S363) #9344 and total RSK #9355, pRSK2 S227 #3556S and total RSK2 (#5528S) (all from Cell Signaling Technology, and used a 1:1,000 dilution), and p-TSC2 S1365 (mouse) #120718 (NovoPro Labs, 1:500). Total protein was determined with Revert Total Protein Stain (926-11016; LI-COR). TSC2 S1364A and S1364E mutants (human TSC2) both plasmid and adenovirus were generated as previously described (Ranek et al, 2019). PKG1α C42S–mutant plasmid was generated as previously described (Oeing et al, 2020).

### Stimulation protocols and kinase modulation

In studies using NRVMs, cells were stimulated with endothelin-1 (ET-1 100 nM for 15 min). MEFs were stimulated with either thrombin (100 nM thrombin for 15 min), insulin (0.5 $\mu$M for 30 min), or platelet-derived growth factor (PDGF, 20 ng/ml for 30 min), all dissolved in distilled water. In these studies, cells were further pretreated with ERK1/2 inhibition (SCH772984 or U0126 [both 10 $\mu$M]) and Akt inhibition (MK-2206, 150 nM, unless otherwise indicated) each provided 1 h before stimulation. These doses and incubation times were based on screening assays for inducing Akt, ERK1/2, and S6K activation. In another experiment, NRVMs were pre-incubated with phosphodiesterase type 9 (PF-04449613, 0.1 $\mu$M), type 5 (sildenafil, 1 $\mu$M), or guanylate cyclase activator (Bay-602770, 0.1 $\mu$M). In many of the studies, cells were pre-transfected (plasmid) or infected (adenovirus) to express TSC2 WT, TSC2$^{S1364A}$, or TSC2$^{S136E}$ mutants, at 10 MOI. After 48 h in culture, cells were placed in serum-free media (DMEM) overnight and stimulated with activators (ET-1 [100 nM], insulin [0.5 $\mu$M], PDGF [20 ng/ml], or thrombin [100 nM]) for 15 (ET-1, thrombin) or 30 (insulin, PDGF) minutes.

Simulated ischemia, serum withdrawal, and amino acid depletion/repletion were studied in NRVMs. To simulate ischemia, NRVMs were subjected to 30 min of hypoxia, acidosis, and reduced glucose using a hypoxia chamber and ischemia-mimicking buffer containing 125 mM NaCl, 6.25 mM NaHCO$_3$, 1.2 mM KH$_2$PO$_4$, 1.2 mM CaCl$_2$, 1.25 mM MgSO$_4$, 5 mM sodium lactate, 8 mM KCl, 20 mM Hepes, and 20 mM 2-deoxyglucose (pH adjusted to 6.6) as described (Oeing et al, 2021). For serum withdrawal, cells were provided fresh DMEM supplemented with 10% FBS for 1 h followed by incubation with DMEM lacking FBS for 1 h. Amino acid depletion/repletion used DMEM deplete of amino acids and supplemented with 10% dialyzed FBS and 1% pen/strep for 1 h, followed by medium containing the full complement of amino acids, DMEM, and supplemented with 10% dialyzed FBS and 1% pen/strep for 1 h.

### PKG-dependent TSC2 phosphorylation

To assess intrinsic TSC2 S1364 phosphorylation, NRCMs were cultured in no-serum media for 24 h before study, then pretreated with SCH772984 (10 $\mu$M), Akt (MK-2206, 150 nM), PKG (DT3, 1 $\mu$M), or vehicle for 1 h before ET-1 (100 nM), insulin (10 $\mu$g/ml), or vehicle for 15 min. Cells were then lysed and assayed for TSC2 S1364 phosphorylation.

### Immunoblotting

Cells were lysed in lysis buffer (9803; Cell Signaling Technology) and protein concentrations determined by BCA assay (Pierce). Samples were prepared in Licor protein sample loading buffer (LI-COR) or SDS Tris-glycine buffer (Cat no. -1610772; Life Technologies) and run on TGX 7.5% or 4–20% Tris-glycine gels (Cat. no. 456-1026, 5671025; Bio-Rad) and blotted onto nitrocellulose membranes. The membranes were blocked in blocking buffer (927-60001; LI-COR), and incubated with primary antibodies diluted in antibody buffer (Cat. no. -927-65001; LI-COR) at 4°C overnight. A total protein stain (Cat. no. -926-11021; LI-COR, Lot #-D10129-02) was used at 5–10 ml per membrane. Antibody binding was visualized with an infrared imaging system (Odyssey; LI-COR) and bands were quantified with Odyssey Application Software 3.1.

### In vivo high-fat diet–induced obesity

Age matched male mice harboring homozygous KI mutations for SA (n = 8), SE (n = 14) or littermate WT controls (n = 14) were fed 60%

high fat diet (Research Diets, D12492, 60 kcal% fat) for 18 wk, starting at age 10–12 wk. An additional group of n = 3–5 mice of each genotype were placed on a standard chow diet for the same time. Serial weights were obtained every other week. After 18 wk resting plasma glucose and glucose tolerance tests were obtained after overnight fasting (18 h). For the glucose tolerance test, the mice were injected intraperitoneally with glucose (1 g/kg) and a baseline glucose measurement was taken from a tail-clip blood droplet using a glucometer. Blood glucose was then measured serially at 15, 30, 60, and 120 min. At the end of the protocol, livers were harvested for histology and fixed in SafeFix Fixative. Paraffin-embedded, dewaxed, 5-µm sections were stained using hematoxylin and eosin. The study was approved by the Johns Hopkins Animal and Care Use Committee in accordance with NIH guidelines on the humane care and use of animals.

### Statistical analysis

Data are mostly analyzed by one- or two-way ANOVA or by nonparametric Kruskal–Wallis or Mann–Whitney U tests, as appropriate. Each test is identified in the figure legends. Multiple-comparisons tests (Sidak, Dunn's, or Tukey) were used to obtain within group comparisons, and results are also provided in the legends. Statistical analysis was performed using Prism Version 9.0.

## Data Availability

The study does not report any large-scale data bases, sequences, computation models, or atomic coordinates. We have provided original gels for all data displayed in the manuscript, as well as source data for all summary figures. Each figure is accompanied by a PDF file with the original gels, and an Excel spreadsheet providing the numerical data shown in the summary graphs.

## Supplementary Information

## Acknowledgements

This study was supported by National Institute of Health–Heart Lung and Blood Institute grants: HL135827 (DA Kass) and HL 143905 (BL Dunkerly-Eyring); T32-HL-7227 (M Pinilla-Vera); American Heart Association 16SFRN28620000 (DA Kass) and 18CDA34110140 (MJ Ranek); and Deutsche Forschungsgemeinschaft (German Research Foundation) OE 688/1-1 (CU Oeing), BIH-Charité clinical scientist program funded by the Charité–Universitätsmedizin Berlin and the Berlin Institute of Health (CU Oeing).

### Author Contributions

BL Dunkerly-Eyring: data curation, formal analysis, investigation, and writing—original draft.
S Pan: data curation and formal analysis.
M Pinilla-Vera: formal analysis and investigation.
D McKoy: investigation.
S Mishra: formal analysis and investigation.
MI Grajeda Martinez: investigation.
CU Oeing: formal analysis and investigation.
MJ Ranek: resources and methodology.
DA Kass: conceptualization, data curation, formal analysis, supervision, funding acquisition, investigation, project administration, and writing—review and editing.

### Conflict of Interest Statement

DA Kass, BL Dunkerly-Eyring, and MJ Ranek are co-inventors on a PCT patent regarding the use of TSC2 S1364/65 (human) mutations for immunological therapeutics.

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
