## [Reviewer comments · Life Science Alliance]

Life Science Alliance

Single Serine on TSC2 Exerts Biased Control over mTORC1 Activation by ERK1/2 but Not Akt

Brittany Dunkerly-Eyring, Shi Pan, Miguel Pinilla-Vera, Desirae McKoy, Sumita Mishra, Maria Grajeda Martinez, Christian Oeing, Mark Ranek, and David Kass

DOI: <https://doi.org/10.26508/lsa.202101169>

Corresponding author(s): David Kass, Johns Hopkins University

Review Timeline:

Submission Date:	2021-07-23
Editorial Decision:	2021-08-24
Revision Received:	2022-01-23
Editorial Decision:	2022-02-18
Revision Received:	2022-02-23
Accepted:	2022-02-25

Scientific Editor: Novella Guidi

Transaction Report:

August 24, 2021

Re: Life Science Alliance manuscript #LSA-2021-01169-T

Dr. David A. Kass
Johns Hopkins Univ. School of Medicine
Medicine
720 Rutland Ave., Ross 858
720 Rutland Avenue
Baltimore, MD 21205

Dear Dr. Kass,

Thank you for submitting your manuscript entitled "Single Serine on TSC2 Exerts Biased Control over mTORC1 Activation by ERK1/2 but Not Akt" to Life Science Alliance. The manuscript was assessed by expert reviewers, whose comments are appended to this letter. As you will note from the reviewers' comments below, both reviewers are quite positive about the study and feels that is carefully conducted and that the data and figures are well presented and largely support the authors' conclusion. Reviewer 1 main issues are the text unclarity and the use of ERK1/2 inhibitor SCH772984, which does not seem to inhibit ERK1/2 by altering ERK1/2 phosphorylation, in contrast with other reports. Please also evaluate the effect of thrombin on S6K phosphorylation in the presence of SCH772984 in SE mutant, as suggested by this reviewer. Reviewer 2 main concern is that the evaluation of mTORC1 activity was only conducted by analyzing the phosphorylation of S6K, therefore please evaluate also 4E-BP1, autophagy (ULK1) or lysosomal targets such as TFEB. Also, please provide discussion whether only ERK-mediated S6K phosphorylation is sufficient to change cell metabolism or cell size. We, thus, encourage you to submit a revised version of the manuscript back to LSA that responds to all of the reviewers' points.

Thank you for this interesting contribution to Life Science Alliance. We are looking forward to receiving your revised manuscript.

Sincerely,

-- Summary blurb (enter in submission system): A short text summarizing in a single sentence the study (max. 200 characters including spaces). This text is used in conjunction with the titles of papers, hence should be informative and complementary to

the title and running title. It should describe the context and significance of the findings for a general readership; it should be written in the present tense and refer to the work in the third person. Author names should not be mentioned.

B. MANUSCRIPT ORGANIZATION AND FORMATTING:

Reviewer #1 (Comments to the Authors (Required)):

The manuscript by Dunkerly-Eyring et al. expands on their earlier exciting data showing that the phosphorylation of S1365 of TSC2 potently blocks mTORC1 co-activation by pathological stress, while its dephosphorylation acts the opposite. In the present study, the authors propose an interesting hypothesis that S1365 modification status exerts control over mTORC1 activity via ERK1/2 but not Akt-dependent signalling.

This is a carefully conducted study, the data and figures are well presented and largely support the authors' conclusion. There are a few comments and technical issues that need to be addressed to improve clarity of the manuscript.

Major points:

- (1) By far the weakest area of this manuscript is the text. I can honestly say I didn't understand the summary at all! Most work to date on TSC-mTORC1 signalling is done in response to insulin, so the fact that these authors are able to demonstrate that different stimuli (e.g. ET1, PDGF, thrombin, stimulated ischaemia) show different responses is an important finding for the field. At present however, it is hard to understand what the models are, why they are important and how they fit together. To reach a wider audience, this needs to be made much clearer. The results section is chaotic in places- it is often difficult to figure out, which experiment is actually being referred to and described; some figure references are missing as well. The discussion could also re-written for clarity, especially to make the model presented in Fig. 7 more understandable.
- (2) The authors rely heavily on the use of ERK1/2 inhibitor: SCH772984. Although there are reports showing clearly that this inhibitor blocks ERK1/2 by altering ERK1/2 phosphorylation, the data provided in the manuscript do not demonstrate this effect. Could the authors explain the cause of this phenomenon or provide alternative evidence showing that experimental conditions used in the study were able to potently inhibit ERK1/2 (e.g., by showing RSK2 phosphorylation status, which is an ERK1/2 substrate also inhibited by SCH772984)? Was the effect the same in NRVM and MEF cell lines (e.g. in Fig. 4C the authors do not show ERK1/2 phosphorylation status, even though the experiment conducted on MEFs relied on SCH773984-directed inhibition)?
- (3) In Fig. 4 the authors demonstrate that S1365 does not impact Akt signalling when co-activated with ERK1/2. These results are nice evidence supporting their hypothesis; however, I feel that one of the major experiments is missing - what is the effect of thrombin on S6K phosphorylation in the presence of SCH772984 in SE mutant? This figure also lacks consistency. Fig. 4A shows only p-ERK and p-Akt - why not show p-S6K as well (and a loading control)? Whereas Fig. 4C has a full set of proteins, in Fig. 4B and D, p-Akt and p-ERK are missing. Can the authors also confirm that the analysis of TSC WT in 4B is same or different to 4D? The MK-dependent reduction is thrombin-induced increased pS6K is not very convincing in 4D compared to 4B but the quantification (4F) looks very convincing.

Minor points:

- (1) In the figures, could the authors re-label SE/SA to TSC2S1364E etc. This will make interpreting the figures much easier, without need to keep referring to the text to confirm they are still talking about TSC2
- (2) The labels for cells used in the experiment presented in Fig. 2C are missing (WT or TSC2 SA or C42S?). Also, has this result been shown in their previous paper? It's not clear why it needs to be shown in this manuscript as well.
- (3) The explanation on how the results from Fig. 1B and 2A were combined into one figure (Fig. 2D) should be more detailed - the reviewer is not sure how two separate experiments could be merged into one to match p/t-S6K ratios from different sets of blots.
- (4) Fig. 5 should be consistent; why is there no t-S6K in Fig. 5A and no HA in Fig. 5A/B?
- (5) The information on magnification/scale bars are missing from Fig. 6D.
- (6) The information about the anti-phospho-TSC2 (S1366) antibody used in the study is missing from the Reagents section.
- (7) The section describing the results in mice should be expanded as is not clear how the experiments were conducted, especially the histological analysis (this description is also missing in the Materials and methods section).

Reviewer #2 (Comments to the Authors (Required)):

This work by Dunkerly-Eyring et. al. investigates the impact of the TSC2 serine 1365 phosphorylation on ERK- and AKT-mediated mTORC1 activation. In neonatal rat ventricular myocytes and mouse fibroblasts expressing a TSC2-phosphomimetic S1365E mutant, the authors observed a suppression of ERK-mediated mTORC1 activation. This mutant was already described previously, but its importance for ERK-mediated mTORC1 activation was shown for the first time. They were able to discriminate ERK-mediated mTORC1 activation by endothelin 1 treatment and AKT-mediated mTORC1 activation by PDGF and insulin treatment. Neither in vivo nor in vitro TSC2-S1365E/A mutants show effects on AKT-mediated mTORC1 activation. This diverse effect is interesting, but it is a rather specific effect. It has no effect on a whole body metabolism and HFD treatment of mice expressing the TSC2-S1365E/A mutants develop a similar fatty liver as wildtype mice do. For me, the question remain, whether there is any disease known with this mutation of TSC2.

1. One main point is the evaluation of mTORC1 activity. The authors analyzed only the phosphorylation of S6K, which is a read-out for basal mTORC1 activity. Is there any effect on 4E-BP1, autophagy (ULK1) or lysosomal targets such as TFEB? Time: 1 month

2. As this phosphorylation of TSC2 only reduces ERK-mediated S6K phosphorylation without abolishing it, are these effects sufficient to change e.g. cell metabolism (e.g. ROS formation) or cell size? Time: 2 months

3. The analysis of AKT-mediated mTORC1 activation is done in several systems and sufficient to claim, that there is no influence.

Minor comments:

Some sentences (especially in the summary) would need rephrasing to allow an easier reading flow. In Figure 6, the scale bars are missing and an OilRedO staining for easier assessing the abundance of lipid droplets would be preferable.

Reviewer #1 (Comments to the Authors (Required)):

The manuscript by Dunkerly-Eyring et al. expands on their earlier exciting data showing that the phosphorylation of S1365 of TSC2 potentially blocks mTORC1 co-activation by pathological stress, while its dephosphorylation acts the opposite. In the present study, the authors propose an interesting hypothesis that S1365 modification status exerts control over mTORC1 activity via ERK1/2 but not Akt-dependent signaling.

This is a carefully conducted study, the data and figures are well presented and largely support the authors' conclusion. There are a few comments and technical issues that need to be addressed to improve clarity of the manuscript.

Thank you for these supportive comments.

Major points:

(1) By far the weakest area of this manuscript is the text. I can honestly say I didn't understand the summary at all! Most work to date on Tsc-mTORC1 signaling is done in response to insulin, so the fact that these authors are able to demonstrate that different stimuli (e.g. ET1, PDGF, thrombin, stimulated ischaemia) show different responses is an important finding for the field. At present however, it is hard to understand what the models are, why they are important and how they fit together. To reach a wider audience, this needs to be made much clearer. The results section is chaotic in places- it is often difficult to figure out, which experiment is actually being referred to and described; some figure references are missing as well. The discussion could also be rewritten for clarity, especially to make the model presented in Fig. 7 more understandable.

We are sorry the text was found confusing. The various mTORC1 stimulation methods we employed (e.g. ET-1, thrombin, insulin, PDGF, nutrient depletion) have been used before and most involve TSC2 in one manner or another as a signaling nexus. Our focus here was on whether another TSC2 control pathway modulated by TSC2 S1364 PTMs regulates these other modulators in a broad or selective manner. That this one serine confers potent co-modulation of one input kinase (ERK1/2) yet not another (Akt) is to our knowledge, quite novel, and we hope this central message now gets through better in the revision.

(2) The authors rely heavily on the use of ERK1/2 inhibitor: SCH772984. Although there are reports showing clearly that this inhibitor blocks ERK1/2 by altering ERK1/2 phosphorylation, the data provided in the manuscript do not demonstrate this effect. Could the authors explain the cause of this phenomenon or provide alternative evidence showing that experimental conditions used in the study were able to potentially inhibit ERK1/2 (e.g., by showing RSK2 phosphorylation status, which is an ERK1/2 substrate also inhibited by SCH772984)? Was the effect the same in NRVM and MEF cell lines (e.g. in

Fig. 4C the authors do not show ERK1/2 phosphorylation status, even though the experiment conducted on MEFs relied on SCH773984-directed inhibition)?

Thank you for the comment. We agree there are literature showing reduced p/t ERK1/2 with SCH772984 in other cell types, though we did not find prior data specifically for cardiomyocytes stimulated by ET1. That stated, we agree it is essential to confirm ERK1/2 was indeed inhibited, so we have repeated the experiment and performed new assays for phospho/total RSK2 as you suggested. These do confirm a significant reduction with SCH773984 in a dose dependent manner that parallels decline in p/t S6K. Data are now added to revised Figure 1A, and discussed. We also tested a second inhibitor (U-0126) that is more of a direct ERK1/2 inhibitor, and we find this also reduces ET-1 mediated S6K phosphorylation as well as p/t ERK1/2 and RSK p/t. These data are now provided in a new Supplemental Figure 1A. We also repeated the study shown in Figure 1C where the TSC2 S1364E mutation was introduced to depress p/t S6K. In this instance, the mechanism is downstream of ERK1/2 and we indeed find RSK p/t remains elevated even as p/t S6K declines. We also tested U-0126 and SCH in this study as positive controls, and both result in a decline in RSK activation. This is now provided in a new Supplemental Figure 1B.

(3) In Fig. 4 the authors demonstrate that S1365 does not impact Akt signaling when co-activated with ERK1/2. These results are nice evidence supporting their hypothesis; however, I feel that one of the major experiments is missing - what is the effect of thrombin on S6K phosphorylation in the presence of SCH772984 in SE mutant? This figure also lacks consistency. Fig. 4A shows only p-ERK and p-Akt - why not show p-S6K as well (and a loading control)? Whereas Fig. 4C has a full set of proteins, in Fig. 4B and D, p-Akt and p-ERK are missing. Can the authors also confirm that the analysis of TSC WT in 4B is same or different to 4D? The MK-dependent reduction is thrombin-induced increased pS6K is not very convincing in 4D compared to 4B but the quantification (4F) looks very convincing.

Thank you for the comments. We agree an experiment with the SE mutant and SCH772984 is important, and now provide this using adenovirus gene transfer in TSC2 KO MEFs. The new results (Figure 4G) show that MEFs expressing a WT TSC2 have a greater rise in pS6K from thrombin and larger decline with SCH772984 added when compared to cells expressing the TSC2 S1364E mutant. The interaction of genotype and SCH772984 effect is significant at $p=0.036$. The SE mutation did not prevent a rise in p/t S6K with thrombin stimulation over vehicle control (also $P\leq 0.022$) that we attribute to unblocked Akt.

As far as consistency of presentations, Figure 4A now has a total protein loading control. We did not run p/tS6K for that study as it was done after many others shown in Fig 4 that demonstrated the rise in p/t S6K from thrombin. This particular run was to confirm co-activation of Akt and ERK1/2 pathways by thrombin. We did not have sufficient residual lysates to probe for all of the other signals in Figures 4B and 4D, and felt redoing the entire experiments for this would not impact our primary findings or conclusions. In Figure 4B, we show p-ERK to demonstrate that the Akt inhibitor does not impact ERK phosphorylation, while in Figure 4C, we show p-Akt to demonstrate that the ERK inhibitor does not impact

Akt phosphorylation. Figure 1B had already shown the Akt inhibitor markedly reduces p/tAkt. Figure 4D tests the effect of Akt blockade in cells with the SE mutation, so our primary focus was on pS6K.

Regarding your question about the WT TSC2 experiments in 4B and 4D, they were independently performed, but the way the data had been plot made it look far greater in 4D than 4B due to different y-axis scaling. Our error. The revision shows these data using the same y axis, and the thrombin response and impact of Akt inhibition is very similar.

Minor points:

(1) In the figures, could the authors re-label SE/SA to TSC2S1364E etc. This will make interpreting the figures much easier, without need to keep referring to the text to confirm they are still talking about TSC2

We have done this where feasible given the space needed for the various figure headings, and wanting to maintain readability. With the exception of the PKG mutation presented in Figure 2C – that involves a C42S substitution, the only mutation in this paper is serine to alanine or glutamate at S1364 (human sequence numbering now used throughout the paper for clarity) in TSC2.

(2) The labels for cells used in the experiment presented in Fig. 2C are missing (WT or TSC2 SA or C42S?). Also, has this result been shown in their previous paper? It's not clear why it needs to be shown in this manuscript as well.

These data are all obtained in myocytes expressing the TSC2^{S1364A} (SA) and the PKG1a^{C42S} mutation. This is explained in the results text and figure legend. This particular result – related to ERK1/2 activation (or not) was not previously reported in the earlier paper. In the context of the new findings regarding selective ERK1/2 signaling modulation by the TSC2 S1364A.

(3) The explanation on how the results from Fig. 1B and 2A were combined into one figure (Fig. 2D) should be more detailed - the reviewer is not sure how two separate experiments could be merged into one to match p/t-S6K ratios from different sets of blots.

We have added this detail. Figure 1B and 2C (it was not 2A, that was our error) were used. As presented individually, each data set were already normalized so Vehicle control for each had a mean value of 1.0 for p/t S6K and p/t ERK1/2. What was done further here was to normalize the peak p/t S6K signal as well (arbitrarily set at 4x baseline). This bi-normalization allowed us to combined the different blots into a single analysis as shown.

(4) Fig. 5 should be consistent; why is there no t-S6K in Fig. 5A and no HA in Fig. 5A/B?

The experiment shown in panel 5A was performed by a co-author who at the time did not generate total protein for S6K but used total protein staining to confirm equal loading. Ischemia/hypoxia is not known to essentially deplete total S6K protein, and as the results for pS6K were rather striking, we felt normalization to total protein was adequate. HA had not been probed for the other studies and has been removed for consistency. Total TSC2 protein is shown.

(5) *The information on magnification/scale bars are missing from Fig. 6D.*

We have added magnification bars.

(6) *The information about the anti-phospho-TSC2 (S1365) antibody used in the study is missing from the Reagents section.*

This has been added. Other details regarding all our antibodies (source, catalog number, dilutions) are provided.

(7) *The section describing the results in mice should be expanded as is not clear how the experiments were conducted, especially the histological analysis (this description is also missing in the Materials and methods section).*

The details of the obesity protocol in intact mice is now provided in the revised methods section.

Reviewer #2 (Comments to the Authors (Required)):

This work by Dunkerly-Eyring et. al. investigates the impact of the TSC2 serine 1365 phosphorylation on ERK- and AKT-mediated mTORC1 activation. In neonatal rat ventricular myocytes and mouse fibroblasts expressing a TSC2-phosphomimetic S1365E mutant, the authors observed a suppression of ERK-mediated mTORC1 activation. This mutant was already described previously, but its importance for ERK-mediated mTORC1 activation was shown for the first time. They were able to discriminate ERK-mediated mTORC1 activation by endothelin 1 treatment and AKT-mediated mTORC1 activation by PDGF and insulin treatment. Neither in vivo nor in vitro TSC2-S1365E/A mutants show effects on AKT-mediated mTORC1 activation. This diverse effect is interesting, but it is a rather specific effect. It has no effect on a whole body metabolism and HFD treatment of mice expressing the TSC2-S1365E/A mutants develop a similar fatty liver as wildtype mice do. For me, the question remains, whether there is any disease known with this mutation of TSC2.

The Leiden Open Variation Data Base which lists registries for tuberous sclerosis reports four individuals, 2 each with S1364 or S1365 mutations – 3 of which are felt to be benign, one potentially disease related, though not by itself. For clarity, the mouse TSC2 S1365 corresponds to human S1364. This is consistent with our prior study (Nature 2019) that showed our KI mice with silencing (or mimetic) mutations had negligible rest phenotype. However, in the presence of pressure-overload stress, the TSC2^{S1365A} homozygous KI mice had very high mortality, severe hypertrophy, suppressed autophagy, and amplified mTORC1. All were fully rescued by blocking mTOR with everolimus. The same S1365A mutation was cardio-protective in the setting of cardiac ischemia-reperfusion injury due to metabolic changes (Oeing et al, Circ Res, 2021). Both pressure overload and IR engage ERK1/2 and other MAP kinase activation pathways. The potent regulation of these conditions by TSC2 S1365 mutations is in striking comparison to the negligible impact they have in the current high fat diet-obesity model we report for the first time in the current study. Indeed, when we did the HFD in vivo study, we did not know about the ERK1/2 vs Akt bias, and had been expecting a major impact on insulin signaling and worse metabolic syndrome in the SA KI mutant mice. That nothing had changed puzzled us and led to the cellular work presented in the current study. From the standpoint of mTORC1 regulation, our data reveals a rather dramatic disparity between inter-molecular regulatory controls by one TSC2 serine on others far away from it – depending on exactly which kinases are involved.

1. *One main point is the evaluation of mTORC1 activity. The authors analyzed only the phosphorylation of S6K, which is a read-out for basal mTORC1 activity. Is there any effect on 4E-BP1, autophagy (ULK1) or lysosomal targets such as TFEB? Time: 1 month*

In our 2019 paper, we reported that in myocytes subjected to ET1 or in vivo hearts to pressure-stress, genetically modifying S1365 significantly altered S6K, 4E-BP1 and Ulk1 phosphorylation in similar directions. In our 2021 paper studying ischemia/reperfusion, we also examined all of these, but found the dominant effect in that setting were via pS6K. It is known that various mTORC1 stimuli impact downstream effectors differently though S6K is common to most. That feature was why we focused on this particular readout. Assessing the other three proteins for all our assays while potentially interesting would not change the primary conclusion that ERK1/2 but not Akt-dependent TSC2 regulation is modulated by S1365 with respect to the major mTORC1 regulatory kinase S6K. However, in response to your query, we have provided 4E-BP1 data for conditions where prior data had not been reported, specifically insulin, PDGF, simulated ischemia, and amino acid repletion (new Supplemental Figure 2). Insulin significantly increased p/t 4E-BP1 in WT and SA expressing but not SE expressing cells. There was no change in p/t 4E-BP1 by PDGF stimulation nor impact from the TSC2 mutations, and we observed similar marked declines in this phosphorylation in cells subjected to simulated ischemia or nutrient deficiency. These data are noted where appropriate in the revised results section and gels and summary data presented in Supplemental Figure 2.

2. *As this phosphorylation of TSC2 only reduces ERK-mediated S6K phosphorylation without abolishing it, are these effects sufficient to change e.g. cell metabolism (e.g. ROS formation) or cell size? Time: 2 months*

These are important questions that we addressed in our 2019 study (Ranek et al, Nature 2019; 566, 264–269) in which myocytes were transfected with either TSC2 S1365A, S1365E, or WT protein, and then stimulated with ET-1 for 24 hours. Cell size markedly increased in cells expressing the SA mutation coupled to amplified mTORC1 activation, whereas it declined in those with the SE mutation as did mTORC1 activation. The same was then shown for the intact heart in response to pressure overload (marked hypertrophy with SA mutation and marked suppression of hypertrophy with SE, even in heterozygotes. Our cell studies further showed the mutations potentially regulated autophagy, SA depressing and SE stimulating autophagic flux. In our 2021 study on ischemia reperfusion (Oeing et al, Circ Res 2021; 128:639-651) we showed the SA mutation confers potent changes in myocardial metabolism, favoring glucose over fat oxidation, to afford cardio-protection ex vivo and in vivo. We referred to these data in our paper and they are now more fully discussed in the revised text.

3. *The analysis of AKT-mediated mTORC1 activation is done in several systems and sufficient to claim, that there is no influence.*

Thank you for the comment.

Minor comments:

4. *Some sentences (especially in the summary) would need rephrasing to allow an easier reading flow. In Figure 6, the scale bars are missing and an OilRedO staining for easier assessing the abundancy of lipid droplets would be preferable.*

Summary and Discussion has been rewritten for clarity. We have added scale bars for Fig 6. While lipid can be more strikingly stained with OilRedO, it was clearly present by H/E stain, and the lack of impact of the mutation equally clear. Thus, we did not redo this analysis.

February 18, 2022

RE: Life Science Alliance Manuscript #LSA-2021-01169-TR

Dr. David A. Kass
Johns Hopkins University
Medicine
720 Rutland Ave., Ross 858
720 Rutland Avenue
Baltimore, MD 21205

Dear Dr. Kass,

Thank you for submitting your revised manuscript entitled "Single Serine on TSC2 Exerts Biased Control over mTORC1 Activation by ERK1/2 but Not Akt". We would be happy to publish your paper in Life Science Alliance pending final revisions necessary to meet our formatting guidelines.

- Please address the final remaining Reviewer 2 points
- please upload all figure files as individual ones, including the supplementary figure files; all figure legends should only appear in the main manuscript file
- please add your main and supplementary figure legends to the main manuscript text after the references section
- please add ORCID ID for the corresponding author-you should have received instructions on how to do so
- please upload your main manuscript text as an editable doc file
- please note that titles in the system and manuscript file must match
- please make sure the author order in your manuscript and our system match
- please consult our manuscript preparation guidelines <https://www.life-science-alliance.org/manuscript-prep> and make sure your manuscript sections are in the correct order and labeled correctly
- please use the [10 author names, et al.] format in your references (i.e. limit the author names to the first 10)
- please add Data Availability section and approval statement for the mice experiment.
- blots in figure 1B in the row P-akt the second part are hardly visible. Please provide a more exposed version of it and source data for this figure.
- in blots in figure 5C there are white dots that look like water drops

A. FINAL FILES:

-- Summary blurb (enter in submission system): A short text summarizing in a single sentence the study (max. 200 characters including spaces). This text is used in conjunction with the titles of papers, hence should be informative and complementary to

the title. It should describe the context and significance of the findings for a general readership; it should be written in the present tense and refer to the work in the third person. Author names should not be mentioned.

B. MANUSCRIPT ORGANIZATION AND FORMATTING:

Sincerely,

Reviewer #1 (Comments to the Authors (Required)):

In the revised manuscript Dunkerly-Eyring and co-authors addressed the points raised by this reviewer. The authors provide further evidence for control over mTORC1 activity via S1365 modification status through ERK1/2 signalling by adding a few essential controls as well as additional experiments. The authors carefully improved the introduction, results, and discussion sections, which make the text clearer and should make it easier to understand by wider audience.

Overall, I am satisfied with the response and I think it will be a nice addition to the field.

Reviewer #2 (Comments to the Authors (Required)):

The manuscript text improved tremendously. Now, the conclusion and the experiments are much better described and easier to follow. To improve the readability even more, the long sentences on page 6 (first paragraph) and 8 (first paragraph) would need rewriting. In the figure legends the used statistical tests should also be described in a similar manner (2-way ANOVA/2WANOVA/ANOVA?).

Overall, all my concerns are satisfyingly addressed either by experiments or explanations of previous work.

February 25, 2022

RE: Life Science Alliance Manuscript #LSA-2021-01169-TRR

Dr. David A. Kass
Johns Hopkins University
Medicine
720 Rutland Ave., Ross 858
720 Rutland Avenue
Baltimore, MD 21205

Dear Dr. Kass,

Thank you for submitting your Research Article entitled "Single Serine on TSC2 Exerts Biased Control over mTORC1 Activation by ERK1/2 but Not Akt". It is a pleasure to let you know that your manuscript is now accepted for publication in Life Science Alliance. Congratulations on this interesting work.

DISTRIBUTION OF MATERIALS:

Again, congratulations on a very nice paper. I hope you found the review process to be constructive and are pleased with how the manuscript was handled editorially. We look forward to future exciting submissions from your lab.

Sincerely,
